# Spoofing Detection and Mitigation in a Multi-correlator GPS Receiver Based on the Maximum Likelihood Principle

**DOI:** 10.3390/s19010037

**Published:** 2018-12-22

**Authors:** Yanbing Guo, Lingjuan Miao, Xi Zhang

**Affiliations:** School of Automation, Beijing Institute of Technology (BIT), Beijing 100081, China; 20080089@bit.edu.cn (Y.G.); miaolingjuan@bit.edu.cn (L.M.)

**Keywords:** repeater, anti-spoofing, ML estimation, multi-correlators, detecting and removing the counterfeit signals

## Abstract

As a structural interference, spoofing is difficult to detect by the target receiver while the advent of a repeater makes the implementation of spoofing much easier. Most existing anti-spoofing methods are merely capable of detecting the spoofing, i.e., they cannot effectively remove counterfeit signals. Therefore, based on the similarities between multipath and spoofing, the feasibility of applying multipath mitigation methods to anti-spoofing is first analyzed in this paper. We then propose a novel algorithm based on maximum likelihood (ML) estimation to resolve this problem. The tracking channels with multi-correlators are constructed and a set of corresponding steps of detecting and removing the counterfeit signals is designed to ensure that the receiver locks the authentic signals in the presence of spoofing. Finally, the spoofing is successfully executed with a software receiver and the saved intermediate frequency (IF) signals, on this basis, the effectiveness of the proposed algorithm is verified by experiments.

## 1. Introduction

GPS is susceptible to a variety of interferences due to the low signal power. For GPS receivers, there are two major types of interferences at present: jamming and spoofing. Strong interferences are used in jamming, usually resulting in loss of lock and positioning failure of the target receiver. Hence the concealment of this interference is poor. On the other hand, spoofing is a structural interference, the counterfeit signals rebroadcasted by spoofers are very similar to the authentic signals from satellites and can gradually induce the positioning results of the target receiver to a false location [1,2,3,4]. Compared with jamming, spoofing is less likely to be found by the target receiver and hence is more sinister.

The early spoofing is simplistic. Specifically, one simply attaches a power amplifier and a transmitting antenna to a satellite signal simulator and broadcasts the counterfeit signals toward the target receiver. However, the simplistic spoofing usually needs the cooperation of jamming to bring the target receiver into the reacquisition first. In 2008, Humphreys et al. successfully developed a repeater based on a software receiver, which can directly deceive the target receiver at the tracking stage. It reminds people of the threat of spoofing once again [1]. Correspondingly, research on anti-spoofing began to develop rapidly. In 2012, Jafarnia-Jahromi et al. made a comprehensive study of spoofing threats and briefly introduced different techniques for two main categories, namely spoofing detection and spoofing mitigation [5]. According to the description of related papers, common spoofing countermeasures can be categorized to four groups: signal power anomaly detection [2,3], time-of-arrival anomaly detection [6,7,8], space processing [9,10,11,12], and correlation peak distortion detection [13,14,15,16,17]. Daneshmand [12] proposed adaptive antenna array beamforming null steering method, which can directly shield the counterfeit signals from a single interference source. However, it is difficult in deploying multiple antennas. In view of the diversity of spoofing, Broumandan used multiple anti-spoofing methods simultaneously on the receiver and achieved the mitigation of simplistic spoofing [18]. However, on the whole, most of the current anti-spoofing techniques are merely able to detect the spoofing. This means that the target receiver can ensure that the positioning results are not affected by closing the corresponding channels after detecting the spoofing only when the number of the authentic signals spoofed with the counterfeit signals is small.

Based on the discussions above, the spoofing mitigation needs to be further researched in anti-spoofing. Multipath and spoofing share the similarity that they distort the correlation peaks of the composite signals. In particular, if the carrier frequency of the counterfeit signal deviates from that of its corresponding authentic signal, the distortion of the correlation peak of the composite signal will be serious. Instead, if the carrier frequency of the counterfeit signal is consistent with that of its corresponding authentic signal, the correlation peak of the composite signal of spoofing will be similar to that of multipath, which is the most difficult situation to detect and handle. Hence, the carrier frequencies of the counterfeit signal and its corresponding authentic signal are set to be the same (frequency lock mode) [19]. As a result, multipath mitigation approaches can be applied to anti-spoofing [20]. However, the counterfeit signals rebroadcasted by spoofers and the multipath signals are significantly different. The major differences are as follows [21]:(1)Except for the special case that the direct signal is blocked, multipath signals from a satellite are typically weaker than its direct signal, while the counterfeit signals for spoofing are usually slightly higher in power than the authentic signals.(2)The correlation peaks of multipath signals lag behind the correlation peak of the corresponding direct signal. The distance between these two kinds of peaks is commonly considered to be quite close. The effective overlapping of these two kinds of peaks indicates a quite close distance. By contrast, the counterfeit signals for spoofing can be aligned with the authentic signals, even ahead of the authentic signals. The distance between the correlation peaks of these two kinds of signals may be either far or close.

It means that not all multipath mitigation methods can be generalized to anti-spoofing. For example, the code phase discriminators of the multipath mitigation methods represented by early/late slope technique are specifically designed for the characteristics of multipath effects and cannot be applied to anti-spoofing. Multipath estimation is another type of important multipath mitigation methods. The composite signals are corrected to restore the direct signals on the basis of estimating the specific parameters of multipath signals. Such methods are usually based on least squares estimation, ML estimation, or Bayesian estimation and require fewer assumptions on multipath signals. Hence, such methods are promising in anti-spoofing.

The rest of this paper is organized as follows. Section 2 briefly introduces the characteristics of a repeater and the spoofing rebroadcasted by it. Section 3 starts from the concepts of ML estimation and constructs the tracking channels with multi-correlators. Section 4 designs a set of corresponding steps of detecting and removing the counterfeit signals. Section 5 simulates the spoofing and verifies the effectiveness of the proposed algorithm. Finally, the work in this paper is summarized in Section 6.

## 2. A Repeater and the Spoofing Rebroadcasted by It

In most cases, the counterfeit signals generated by a satellite signal simulator are not overlapped effectively with the authentic signals present in the space in which the receiver is located. If the counterfeit signal and the authentic signal correspond to the same satellite, and the code phase difference between them is less than 1+d chips, then the overlapping of the two signals is considered to be effective in this paper, where d is the spacing between early and prompt correlators (or the spacing between prompt and late correlators) of the code loop. The autocorrelation and cross-correlation properties of C/A codes mean that the counterfeit signal is equivalent to the noise for the receiver channel which locks the authentic signal, i.e., it is difficult for the simplistic spoofing to affect the positioning results of the tracking receiver. By contrast, the spoofing executed by the repeater can affect the positioning results of the target receiver without reacquisition. As shown in Figure 1, the repeater consists of two modules, namely the receiver module and the spoofer module. The receiver module is the same with an ordinary GPS receiver. The spoofer module properly delays and amplifies the received authentic signals and rebroadcasts the resulting counterfeit signals toward the target receiver. In order to ensure that counterfeit signals are effectively overlapped with the authentic signals, the repeater needs to know the approximate location of the target receiver to determine the appropriate time delay. Since the size of the repeater is usually small, it can be inconspicuously deployed near the target receiver, which facilitates the execution of spoofing [1]. Since it is easy for the simplistic spoofing to be handled, the spoofing mentioned subsequently in this paper is always executed by the repeater and has the form shown in Figure 2. The correlation peaks of the counterfeit signal and the authentic signal have successively gone through three stages: approaching, effective overlapping, and moving away. The amplitude of the counterfeit signal is higher than that of the authentic signal throughout the process. In fact, as long as λ is greater than 1.1, the target receiver will gradually lose its lock on the authentic signal [16]. λ is the power ratio of the counterfeit signal to the authentic signal.

## 3. Construction of Multi-Correlators and Maximum Likelihood Rstimation of Dignal Parameters

In order to achieve the mitigation of spoofing, we employ the idea of multipath estimation. Firstly, in the presence of spoofing, the composite signals for the in-phase (I) and quadrature (Q) branches of any channel of the target receiver can be modeled as
(1)$$i_{\Sigma }\left(t\right)=\sum_{n=0}^{N}a_{n}\left(t\right)A_{0}D\left[t-\tau_{n}\left(t\right)\right]c\left[t-\tau_{n}\left(t\right)\right]\cos\left[\phi_{n}\left(t\right)\right]+v_{i}\left(t\right),$$
(2)$$q_{\Sigma }\left(t\right)=\sum_{n=0}^{N}a_{n}\left(t\right)A_{0}D\left[t-\tau_{n}\left(t\right)\right]c\left[t-\tau_{n}\left(t\right)\right]\sin\left[\phi_{n}\left(t\right)\right]+v_{q}\left(t\right),$$
respectively, where $A_{0}$ is the known amplitude of the authentic signal, D(⋅) is the navigation data bit, c(⋅) is the CA code, N is the number of counterfeit signals in the current channel, n=0,1,2,⋯,N, $v_{i}\left(t\right)$ and $v_{q}\left(t\right)$ denote the noises in the I and Q branches, respectively. $a_{n}\left(t\right)$, $\tau_{n}\left(t\right)$, $\phi_{n}\left(t\right)$ are the amplification coefficient of the amplitude, code delay (time delay) and carrier phase corresponding to the *n*-th signal, respectively. If n=0, they correspond to the authentic signal and $a_{0}\left(t\right)=1,\;\tau_{0}\left(t\right)=0,\;\phi_{0}\left(t\right)=0$. Otherwise, they correspond to the counterfeit signals and $a_{n}\left(t\right)\geq \sqrt{1.1}$. In reality, the repeater delays and amplifies the authentic signal, and then rebroadcasts it. This principle determines that there is usually only one counterfeit signal corresponding to each satellite, namely, N=1. Although the counterfeit signals for spoofing can be ahead of the authentic signal, we assume $\tau_{n}\left(t\right)\geq 0$ by symmetry in order to facilitate the discussion.

As shown in Figure 3, a series of correlators whose time delays are $\beta_{0},\beta_{1},\beta_{2},\cdots ,\beta_{M}$ is deployed in the channel. Take the I branch for example, the composite signal is correlated with these correlators. It is assumed that the variations of $a_{n}\left(t\right)$, $\tau_{n}\left(t\right)$ and $\phi_{n}\left(t\right)$ are negligible and D(⋅) does not flip in the coherent integration time of $T_{\mathrm{coh}}$. Then the result of this coherent integration can be expressed as
(3)$$\begin{array}{cc} I_{m,k} & =\sum_{n=0}^{N}\frac{x_{n}}{T_{\mathrm{coh}}}\int_{t_{k}}^{t_{k}+T_{\mathrm{coh}}}c\left(t-\tau_{n}\right)c\left(t-\beta_{m}\right)dt+I_{m,k}^{v}\\ & =\sum_{n=0}^{N}x_{n}R_{C}\left(\beta_{m}-\tau_{n}\right)+I_{m,k}^{v}, \end{array}$$
where k is the index of the discretized sampling time $t_{k}$, $R_{C}\left(\cdot \right)$ is the autocorrelation function of the CA code, m=0,1,2,⋯,M, and
(4)$$x_{n}=a_{n}A_{0}\cos\left(\phi_{n}\right),$$
(5)$$I_{m,k}^{v}=\frac{1}{T_{\mathrm{coh}}}\int_{t_{k}}^{t_{k}+T_{\mathrm{coh}}}v_{i}\left(t\right)c\left(t-\beta_{m}\right)dt.$$

Similarly, the expressions for the quadrature branch are written as
(6)$$Q_{m,k}=\sum_{n=0}^{N}y_{n}R_{C}\left(\beta_{m}-\tau_{n}\right)+Q_{m,k}^{v},$$
(7)$$y_{n}=a_{n}A_{0}\sin\left(\phi_{n}\right),$$
(8)$$Q_{m,k}^{v}=\frac{1}{T_{\mathrm{coh}}}\int_{t_{k}}^{t_{k}+T_{\mathrm{coh}}}v_{q}\left(t\right)c\left(t-\beta_{m}\right)dt.$$

Assume that $v_{i}\left(t\right)$ and $v_{q}\left(t\right)$ are white Gaussian noises with zero mean and variance $\sigma_{v}^{2}$, then $I_{m,k}^{v}$ and $Q_{m,k}^{v}$ are white Gaussian noises with
(9)$$\left\{ \begin{array}{l} E\left(I_{m,k}^{v}\right)=E\left(Q_{m,k}^{v}\right)=0\\ E\left[I_{m_{1},k}^{v}I_{m_{2},k}^{v}\right]=E\left[Q_{m_{1},k}^{v}Q_{m_{2},k}^{v}\right]=\frac{\sigma_{v}^{2}}{N_{\mathrm{coh}}}R_{C}\left(\beta_{m_{1}}-\beta_{m_{2}}\right), \end{array}\right.$$
where $m_{1}=0,1,2,\cdots ,M$, $m_{2}=0,1,2,\cdots ,M$.

Take the I branch for instance and ignore k. Equation (3) can be extended to a matrix form as
(10)$$I_{\Sigma }=G_{\Sigma }X_{\Sigma }+I_{\Sigma }^{v},$$
where
(11)$$I_{\Sigma }={\left[I_{0}\;I_{1}\;I_{2}\;\cdots \;I_{M}\right]}^{T},$$
(12)$$X_{\Sigma }={\left[x_{0}\;x_{1}\;x_{2}\;\cdots \;x_{N}\right]}^{T},$$
(13)$$I_{\Sigma }^{v}={\left[I_{0}^{v}\;I_{1}^{v}\;I_{2}^{v}\;\cdots \;I_{M}^{v}\right]}^{T},$$
(14)$$G_{\Sigma }=\left[\begin{array}{cccc} R_{C}\left(\beta_{0}-\tau_{0}\right) & R_{C}\left(\beta_{0}-\tau_{1}\right) & \cdots & R_{C}\left(\beta_{0}-\tau_{N}\right)\\ R_{C}\left(\beta_{1}-\tau_{0}\right) & R_{C}\left(\beta_{1}-\tau_{1}\right) & \cdots & R_{C}\left(\beta_{1}-\tau_{N}\right)\\ \vdots & \vdots & \ddots & \vdots \\ R_{C}\left(\beta_{M}-\tau_{0}\right) & R_{C}\left(\beta_{M}-\tau_{1}\right) & \cdots & R_{C}\left(\beta_{M}-\tau_{N}\right) \end{array}\right],$$
and the covariance matrix of $I_{\Sigma }^{v}$ is $P_{\Sigma }=\frac{\sigma_{v}^{2}}{N_{\mathrm{coh}}}C_{\Sigma }$, where
(15)$$C_{\Sigma }=\left[\begin{array}{cccc} 1 & R_{C}\left(\beta_{0}-\beta_{1}\right) & \cdots & R_{C}\left(\beta_{0}-\beta_{M}\right)\\ R_{C}\left(\beta_{1}-\beta_{0}\right) & 1 & \cdots & R_{C}\left(\beta_{1}-\beta_{M}\right)\\ \vdots & \vdots & \ddots & \vdots \\ R_{C}\left(\beta_{M}-\beta_{0}\right) & R_{C}\left(\beta_{M}-\beta_{1}\right) & \cdots & 1 \end{array}\right].$$

It is handy that the observation vector $I_{\Sigma }$ follows a Gauss distribution, namely $I_{\Sigma }∼N\left(G_{\Sigma }X_{\Sigma },P_{\Sigma }\right)$.

It can be assumed that $\tau_{n}$ is known to facilitate the estimation of signal parameters. In this case, only linear problems need to be solved. Actually, $\tau_{n}$ can hardly be consistent with our assumption. In this case, some additional strategies are needed to estimate $\tau_{n}$. See Step 3 in Section 4 for more details. The probability distribution function (PDF) of $I_{\Sigma }$ given $X_{\Sigma }$ can be expressed as
(16)$$f\left(I_{\Sigma }|X_{\Sigma }\right)=\frac{1}{{\left(\sqrt{2\pi }\right)}^{M+1}\det{\left(P_{\Sigma }\right)}^{1/2}}\exp\left\{-\frac{1}{2}{\left[I_{\Sigma }-G_{\Sigma }X_{\Sigma }\right]}^{T}P_{\Sigma }^{-1}\left[I_{\Sigma }-G_{\Sigma }X_{\Sigma }\right]\right\}.$$

Furthermore, we select the following ML function
(17)$$L\left(I_{\Sigma },X_{\Sigma }\right)=\ln f\left(I_{\Sigma }|X_{\Sigma }\right).$$

Then, the ML estimate of $X_{\Sigma }$ [22] is
(18)$${\hat{X}}_{\Sigma }={\left(G_{\Sigma }^{T}C_{\Sigma }^{-1}G_{\Sigma }\right)}^{-1}G_{\Sigma }^{T}C_{\Sigma }^{-1}I_{\Sigma }.$$

Similarly, the ML estimate of $Y_{\Sigma }={\left[y_{0}\;y_{1}\;y_{2}\;\cdots \;y_{N}\right]}^{T}$ for the quadrature branch can be obtained as
(19)$${\hat{Y}}_{\Sigma }={\left(G_{\Sigma }^{T}C_{\Sigma }^{-1}G_{\Sigma }\right)}^{-1}G_{\Sigma }^{T}C_{\Sigma }^{-1}Q_{\Sigma },$$
where $Q_{\Sigma }={\left[Q_{0}\;Q_{1}\;Q_{2}\;\cdots \;Q_{M}\right]}^{T}$.

It can be drawn from the above derivations that M must be no less than N. In particular, it is possible that M=N and $\beta_{0},\beta_{1},\beta_{2},\cdots ,\beta_{M}$ are equal to $\tau_{0},\tau_{1},\tau_{2},\cdots ,\tau_{N}$, respectively. In this setting, Equations (18) and (19) can be simplified as
(20)$${\hat{X}}_{\Sigma }=C_{\Sigma }^{-1}I_{\Sigma },$$
(21)$${\hat{Y}}_{\Sigma }=C_{\Sigma }^{-1}Q_{\Sigma }.$$

When the time delay of a counterfeit signal equals $\beta_{n}$, only the sum of squares of ${\hat{x}}_{n}$ and ${\hat{y}}_{n}$ is not obviously close to 0 regardless of the authentic signal. Other cases and the specific deployment of multiple-correlators will be explained in detail in the next section.

## 4. Detection and Removal of Counterfeit Signals

Since a wide correlator with 0.5 chips spacing between early and prompt correlators (or the spacing between prompt and late correlators) of the code loop is vulnerable to multipath effects or similar spoofing, the spacing d is typically set to 0.5 chips to validate the effectiveness of the proposed anti-spoofing method. It is assumed that the target receiver has locked authentic signals before spoofing is applied. It is well-known that the carrier frequency difference between the counterfeit signal and its corresponding authentic signal makes spoofing very easy to be detected. Since the repeater and the target receiver are usually very close, the repeater obtains the carrier frequency of the authentic signal before delaying and amplifying the authentic signal. As a result, the carrier frequency of the counterfeit signal rebroadcasted by the repeater is close to that of the authentic signal received by the target receiver. Hence, the carrier frequencies of the counterfeit signal and its corresponding authentic signal are set to be the same [19]. The counterfeit signal is effectively overlapped with the authentic signal only when the distance between the correlation peaks of the two signals is less than 1.5 chips. If the distance between the correlation peaks of the counterfeit signal and the authentic signal is greater than 1.5 chips, then the counterfeit signal is considered to have a long delay. This kind of counterfeit signal cannot directly affect the result of the code phase discriminator, but the partial energy of its correlation peak may stimulate a number of correlators with long time delays in the multi-correlator structure. Accordingly, this will adversely affect the estimation of signal parameters and lead to erroneous restoring of the authentic signal. Therefore, the multi-correlator structure should have self-checking capabilities, i.e., it should be able to detect whether the counterfeit signal has a long delay to help the receiver to take reasonable countermeasures in the face of spoofing at different stages (see Figure 2). For these reasons, we let N=8 and note that $\beta_{0},\beta_{1},\beta_{2},\cdots ,\beta_{M}$ are equal to $\tau_{0},\tau_{1},\tau_{2},\cdots ,\tau_{N}$, respectively. $\beta_{n}$ are set to 0.2n chips, respectively, where n=0,1,2,⋯,N.

The time delays of the counterfeit signals for spoofing are usually not equal to those of the deployed multi-correlator. Detection and removal of counterfeit signals should be carried out according to the following steps.

Step 1: Determine whether the signal amplitude of the *n*-th correlator is valid.

According to Equations (20) and (21), the estimated $x_{n}$ and $y_{n}$ are obtained as ${\hat{x}}_{n}$ and ${\hat{y}}_{n}$ and ${\hat{A}}_{n}=\sqrt{{\hat{x}}_{n}^{2}+{\hat{y}}_{n}^{2}}$ denotes the estimate of $A_{n}=a_{n}A_{0}$. In order to facilitate the discussion, ${\hat{A}}_{n}$ is termed as the signal amplitude of the *n*-th correlator. We choose the value of threshold A based on the carrier-to-noise ratio $C/N_{0}$ of the authentic signal. That is, the value of A is set according to $A_{0}$, which can be obtained from the stimulation of the authentic signal to the first correlator based on the assumption that the target receiver locks authentic signals in advance. If ${\hat{A}}_{n}>A$, ${\hat{A}}_{n}$ is valid. Otherwise, ${\hat{A}}_{n}$ is invalid.

Step 2: Determine whether the counterfeit signal has a long delay.

If the counterfeit signal has a long delay, the correlation peak of the counterfeit signal is closest to that of the correlator with a time delay of 1.6 chips. The stimulating effect on this correlator is also the most significant. Hence, if ${\hat{A}}_{8}>{\hat{A}}_{7}+A$, the counterfeit signal is considered to have a long delay and not to be overlapped effectively with the authentic signal. Then go to Step 6. Otherwise, countinue to Step 3. More details on determining whether the counterfeit signal has a long delay will be further explained in the following experiments.

Step 3: Determine the time delay and amplitude of the counterfeit signal.

In practice, there is usually only one counterfeit signal in any channel. In this case, besides ${\hat{A}}_{0}$, the signal amplitude(s) of one single correlator or two adjacent correlators will be valid. Based on this assumption, if ${\hat{A}}_{0}$, ${\hat{A}}_{n}$ and ${\hat{A}}_{n+1}$ are valid, where n=1,2,⋯,7, then
(22)$${\hat{\tau }}_{x}=\tau_{n}+\frac{{\hat{x}}_{n+1}}{{\hat{x}}_{n}+{\hat{x}}_{n+1}}\left(\tau_{n+1}-\tau_{n}\right):=\tau_{n}+\lambda_{x}\left(\tau_{n+1}-\tau_{n}\right),$$
(23)$${\hat{\tau }}_{y}=\tau_{n}+\frac{{\hat{y}}_{n+1}}{{\hat{y}}_{n}+{\hat{y}}_{n+1}}\left(\tau_{n+1}-\tau_{n}\right):=\tau_{n}+\lambda_{y}\left(\tau_{n+1}-\tau_{n}\right).$$

Generally, ${\hat{\tau }}_{x}\neq {\hat{\tau }}_{y}$, so we have
(24)$$\hat{\tau }=\tau_{n}+\frac{\lambda_{x}\hat{x}+\lambda_{y}\hat{y}}{\hat{x}+\hat{y}}\left(\tau_{n+1}-\tau_{n}\right),$$
(25)$$\hat{A}=\sqrt{{\hat{x}}^{2}+{\hat{y}}^{2}},$$
where $\hat{x}$ and $\hat{y}$ are the estimated amplitudes of the I branch and the Q branch of the counterfeit signal, respectively and
(26)$$\hat{x}={\hat{x}}_{n}+{\hat{x}}_{n+1},$$
(27)$$\hat{y}={\hat{y}}_{n}+{\hat{y}}_{n+1}.$$

If only ${\hat{A}}_{0}$ and ${\hat{A}}_{n}$ are valid, then the estimated time delay $\hat{\tau }$ of the counterfeit signal is equal to 0.2n chips and the estimated amplitude $\hat{A}$ is equal to ${\hat{A}}_{n}$, where n=2,3,⋯,7. If only ${\hat{A}}_{0}$ and ${\hat{A}}_{1}$ are valid, then $\hat{\tau }\leq 0.2$ chips. Eliminating the effect of the authentic signal on the estimated amplitude of the first correlator in the I branch, we have from Equation (26)
(28)$$\hat{x}={\hat{x}}_{0}^{\prime }+{\hat{x}}_{1},$$
where ${\hat{x}}_{0}^{\prime }={\hat{x}}_{0}-A_{0}$. Afterwards, $\hat{\tau }$ and $\hat{A}$ can still be obtained using Equations (24) and (25).

If the above assumptions are not valid, it is indicated that the effect of noise is significant or there is more than one counterfeit signal in the current channel. In this scenario, we can either deploy additional correlators near the correlators whose signal amplitudes are valid to calculate $\hat{\tau }$ and $\hat{A}$, or just skip the calculation of $\hat{\tau }$ and $\hat{A}$ such that the authentic signal is directly restored with the valid signal amplitudes. Since this situation is rare, it will not be detailed here.

Step 4: Determine the carrier phase of the counterfeit signal.

In the process of determining the time delay and amplitude of the counterfeit signal, the calculations of $\hat{x}$ and $\hat{y}$ are involved in different cases. It should be pointed out that the $C/N_{0}$ of the authentic signals used in the experiments are relatively high. Correspondingly, the counterfeit signals whose power is slightly higher than that of their corresponding authentic signals also have relatively high $C/N_{0}$. Therefore, based on $\tan\left(\hat{\phi }\right)=\hat{y}/\hat{x}$, the estimated carrier phase of the counterfeit signal can be obtained in three cases. If $\hat{x}>0$, $\hat{\phi }=\arctan\left(\hat{y}/\hat{x}\right)$. If $\hat{x}<0$, $\hat{\phi }=\pi +\arctan\left(\hat{y}/\hat{x}\right)$. If $\hat{x}=0$, $\hat{\phi }=\mathrm{sgn}\left(\hat{y}\right)\pi /2$.

Step 5: Remove the counterfeit signal.

After detecting the time delay, amplitude and carrier phase of the counterfeit signal, the counterfeit signal is confirmed to have a short delay. The opposite counterfeit signal is added to the composite signal to restore the authentic signal. It is worth noting that the assumption of locking the authentic signal in advance can prevent us from mistakenly removing the authentic signal.

Step 6: Discriminate the phase of the authentic signal.

Discriminate the phase of the authentic signal with the normalized early minus late envelope function and go to the next loop.

The corresponding flow chart is shown in Figure 4.

The whole process of detecting and removing the counterfeit signal is elaborated above. Due to the complexity and variability of deliberate interferences, it is very difficult for one single anti-spoofing method to properly handle all kinds of spoofing. For example, the anti-spoofing method based on the multi-correlator structure can help the tracking receiver to keep the authentic signal locked. However, if spoofing during acquisitionalready exists, this method will be incapable of distinguishing between the authentic and counterfeit signals of the receiver. Therefore, multiple anti-spoofing methods should be adopted in the receiver simultaneously [18]. For instance, if the counterfeit signal corresponding to each satellite comes from a single interference source, the antenna can be moved along any small trajectory. In this case, the relative motion between the receiver and the repeater makes the variation of the carrier frequency of each counterfeit signal exactly the same. If the multi-correlator structure can detect the spoofing at the initial tracking stage, this method can be introduced to help the receiver determine whether the locked signal is the authentic signal. If spoofing with unlocked carrier frequency succeeds in spoofing the target receiver, significant abnormalities in the outputs of phase discriminators will be inevitable [19]. In this case, this method can also be introduced to help the receiver determine whether it is necessary to enter the reacquisition stage. If the carrier frequency difference between the counterfeit signal and its corresponding authentic signal increases to a relatively large value, two distinct correlation peaks will appear at the reacquisition stage. That is, the spoofing degenerates into the simplistic form. In addition, other common anti-spoofing methods can also be used on the receiver to detect the simplistic spoofing earlier or more conveniently and ensure the correctness of the positioning results.

## 5. Experimental Results and Analysis

### 5.1. Spoofing

Since it is prohibited to broadcast spoofing signals in real world, the following method is devised to simulate an intentional attack, as shown in Figure 5. Firstly, a previously stored interval of authentic IF GPS signals with a sampling rate of 38.192 MHz is read. Secondly, these loaded signals are acquired and tracked by a software receiver, and the authentic signal from each satellite is amplified by 1.2 times such that the output is denoted as the counterfeit signal. Thirdly, time delay is applied to the counterfeit signal from each satellite. Lastly, we simulate the composite signals received by the target receiver by adding all original authentic signals and certain time-delayed counterfeit signals.

The visible satellites corresponding to the used IF signals are shown in Figure 6. Fixed delay spoofing and variable delay spoofing are applied to PRN22 and PRN21, respectively. The fixed delay spoofing is executed at 300 ms and the delay is 0.6 chips. The variable delay spoofing is also executed at 300 ms and the delay changes gradually from 0 chips to 2 chips. It should be noted that all time domains where there are spoofing signals are highlighted in the following figures.

Figure 7 shows the outputs of the prompt correlators of PRN22 I and Q branches within 1000 ms. The receiver starts tracking at 0 ms. Since the initial tracking value obtained by the conventional acquisition method is not accurate enough, the carrier phase of the local signal gradually aligns with that of the authentic signal until about 180 ms. The output of the I prompt correlator is stabilized at around 1. After the spoofing is executed, the output of the quadrature prompt correlator suddenly changes, which indicates that the carrier phases of the counterfeit signal and the authentic signal are not the same. However, the counterfeit signal lags behind the authentic signal by only 0.6 chips. Their carrier frequencies are close, and the carrier phase difference between them is almost fixed. Therefore, the carrier phases of the local signal and the composite signal align rapidly, and the output of the quadrature prompt correlator is restored to near zero. At the same time, due to the large amplitude of the composite signal, the output of the I prompt correlator becomes larger. In addition, the noises of the prompt correlators of I and Q branches also become larger after the spoofing is executed, which indicates that spoofing signals might increase the noise level and deteriorate the precision of the estimated coordinates (similar to solar radio emission) [23]. As shown in Figure 8, at the beginning of spoofing, there is a significant jump in the output of the code phase discriminator of PRN22, which is confirmed by the results in Figure 7 and vice versa. The effective overlapping of the counterfeit signal and the authentic signal is equivalent to indirectly magnifying $C/N_{0}$. Consequently, after the receiver locks the composite signal, the accuracy of the code phase discriminator is slightly improved.

Figure 9 shows the outputs of the prompt correlators of PRN21 I and Q branches within 1000 ms. On one hand, since the counterfeit signal corresponding to the PRN 21 has the same carrier phase as the authentic signal, the output of the quadrature prompt correlator remains near 0 throughout spoofing. On the other hand, at the beginning of spoofing, the output of the I prompt correlator has jumped to about 2.2, which is the sum of the amplitudes of the counterfeit signal and the authentic signal. It decreases and finally arrives at the steady state of around 1.2 as the counterfeit signal correlation peak gradually moves away from the authentic signal correlation peak. This indicates that the receiver has turned to lock the counterfeit signal. As shown in Figure 10, the accuracy of the code phase discriminator is improved during the effective overlapping of the counterfeit signal and the authentic signal. Since the signal and noise are both amplified at the same time when generating the counterfeit signal, after the two correlation peaks move away from each other, the accuracy of the code phase discriminator is restored to the level at which the spoofing was not executed.

### 5.2. Anti-Spoofing Effect of the Multi-Correlator Structure

In order to facilitate the discussion, in addition to the above spoofing, spoofing with fixed short delays is applied to PRN18, PRN15, and PRN26, and spoofing with fixed long delays is applied to PRN9 and PRN6. The setups of time delays are shown in Table 1. The counterfeit signal and its corresponding authentic signal from each satellite are set to have the same carrier phase. Figure 11 shows the estimation result of PRN22’s carrier phase when the carrier phases of the counterfeit signal and its corresponding authentic signal are different. Hence, the carrier phases of the counterfeit signal and its corresponding authentic signal from each satellite are set to be the same in order to focus on the estimation of time delay and amplitude. In addition, we let $A=0.2A_{0}$ empirically.

Figure 11 shows $\hat{x}$ and $\hat{y}$ of the counterfeit signal of PRN22. It can be seen that the carrier phase of the counterfeit signal is different from that of the authentic signal, which is consistent with the results in Figure 7 and Figure 8. If the effect of the data bit flipping is disregarded, the carrier phase of the counterfeit signal is approximately −π/2. Since the results of ML estimation at different instants are independent, and its effects at different instants are similar, the estimation results at a certain instant are visualized in order to demonstrate the effect of the multi-correlator structure and to verify the estimation method described conveniently in Section 4. Figure 12 shows all correlators’ outputs of I branches of PRN22, PRN18, PRN15, and PRN26, and corresponding estimated time delays and amplitudes of the counterfeit signals. It is observed that the estimates are consistent with the true values, which means the proposed multi-correlator structure is accurate in estimating the time delays and amplitudes of the counterfeit signals.

Figure 13 shows all correlators’ outputs of I branches of PRN9 and PRN6. Since the time delay of the counterfeit signal of PRN9 is close to 1.6 chips, the stimulating effect on the last correlator in subfigure (a) is significant. The receiver can easily determine that the counterfeit signal has a long delay and directly starts discriminating the phase of the authentic signal. Note that the sizes of ${\hat{x}}_{7}$ and ${\hat{x}}_{8}$ in subfigure (b) are close. However, if the time delay of the counterfeit signal is within the effective estimation interval of the multi-correlator structure from 0 to 1.6 chips, it is impossible that ${\hat{x}}_{n}$ and ${\hat{x}}_{n+1}$ of two adjacent correlators are close in size but are opposite in direction. Therefore, this can help the receiver to recognize the counterfeit signal with a long delay, and then perform phase discrimination directly on the authentic signal.

Figure 14 shows the positioning results after applying all the spoofing and the positioning results after employing the proposed anti-spoofing method in an east–north–up geographic coordinate system. The period of data update is 20 ms. In subfigure (a), the positioning of the target receiver is normal at 0-300 ms. Then, all the spoofing is executed at 300 ms. As a result, the positioning results of the target receiver gradually deviate from the real position and finally arrive at the steady state. In subfigure (b), the positioning of the target receiver remains stable and is not obviously affected by the spoofing, which consolidates the effectiveness of the proposed anti-spoofing method.

## 6. Conclusions

In this paper, several anti-spoofing methods are introduced, and their characteristics are briefly summarized. On this basis, the similarities and differences between spoofing and multipath are analyzed. Improvements of the multipath estimation method are proposed to detect and mitigate spoofing accordingly. Based on ML estimation, the tracking channels with multi-correlators are constructed. At the same time, a set of corresponding steps of detecting and removing the counterfeit signals is designed. The experimental results show that the proposed anti-spoofing method can accurately estimate the time delay, amplitude, and carrier phase of the counterfeit signal with a short delay and can then eliminate it by adding the opposite counterfeit signal. Besides, this method is able to perform self-checking such that it can directly discriminate the phase of the authentic signal when the counterfeit signal has a long delay. In conclusion, the proposed method can ensure that the receiver keeps the authentic signal locked when the correlation peak of the counterfeit signal is overlapped with or moves away from that of the authentic signal and can further guarantee that the receiver is positioned correctly in the presence of spoofing.

## Figures and Tables

**Figure 1 sensors-19-00037-f001:**
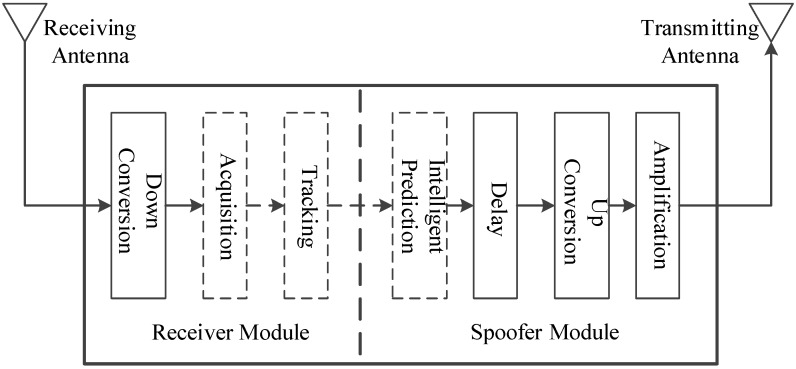
The structure of a repeater.

**Figure 2 sensors-19-00037-f002:**
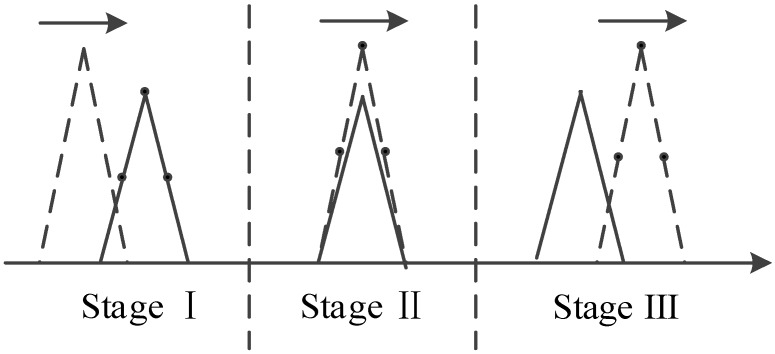
Spoofing rebroadcasted by a repeater.

**Figure 3 sensors-19-00037-f003:**
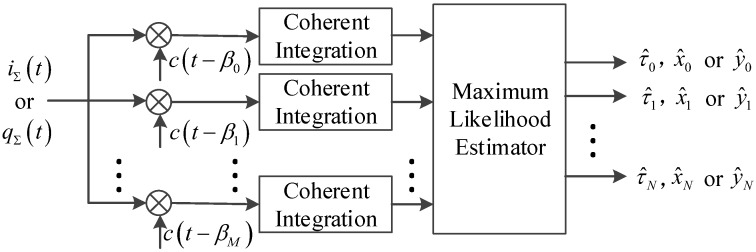
Multi-correlator and maximum likelihood estimator.

**Figure 4 sensors-19-00037-f004:**
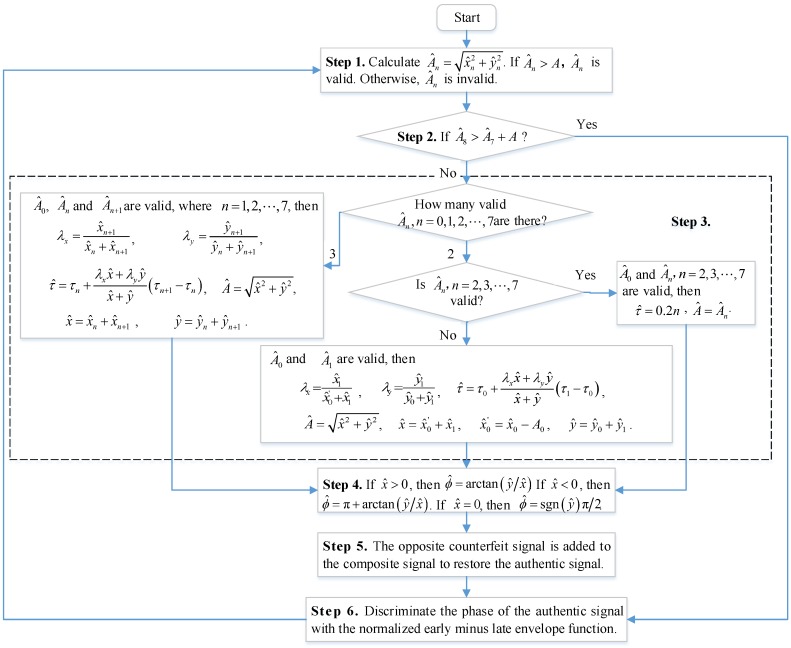
Flow chart for detecting and removing the counterfeit signal.

**Figure 5 sensors-19-00037-f005:**
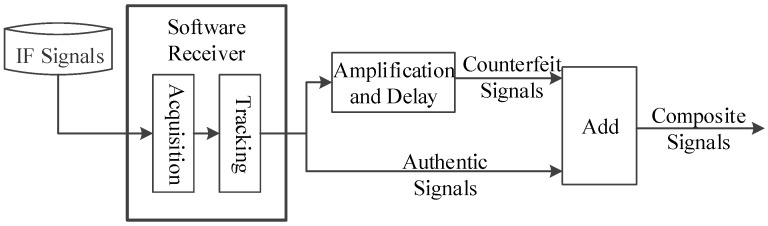
Simulation of spoofing.

**Figure 6 sensors-19-00037-f006:**
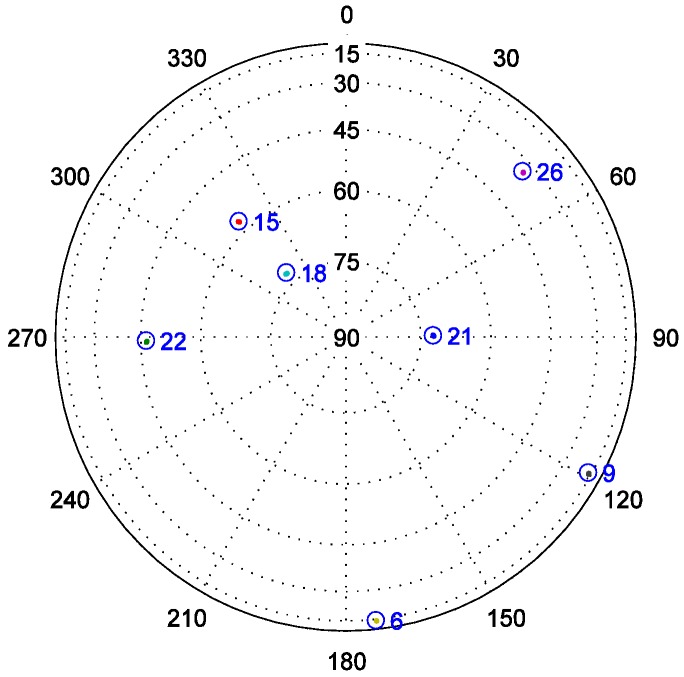
Satellites sky plot.

**Figure 7 sensors-19-00037-f007:**
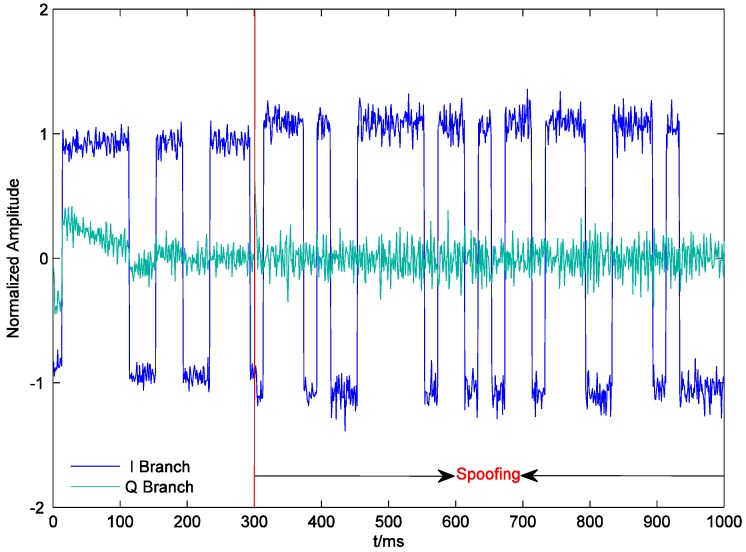
Outputs of the prompt correlators of PRN22 I and Q branches.

**Figure 8 sensors-19-00037-f008:**
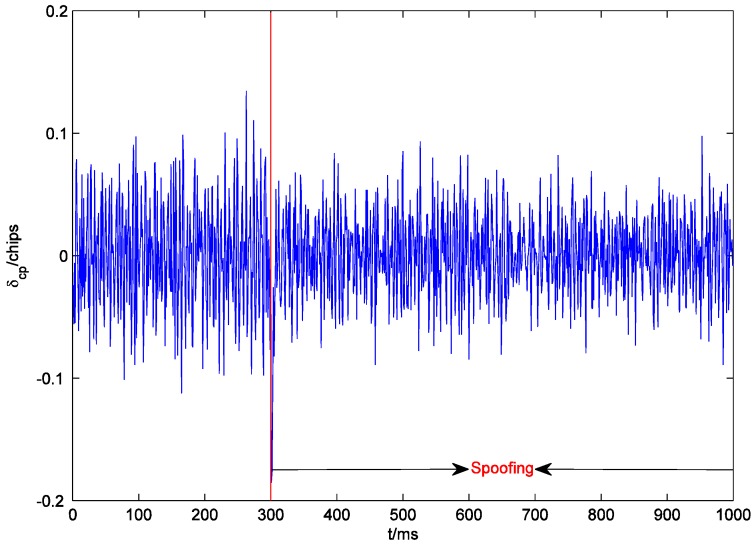
Code phase tracking error $\delta_{\mathrm{cp}}$ of the code phase discriminator of PRN22.

**Figure 9 sensors-19-00037-f009:**
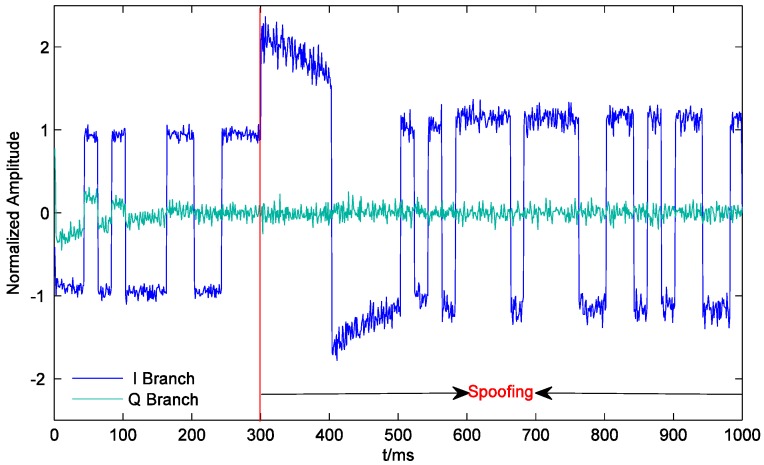
Outputs of the prompt correlators of PRN21 I and Q branches.

**Figure 10 sensors-19-00037-f010:**
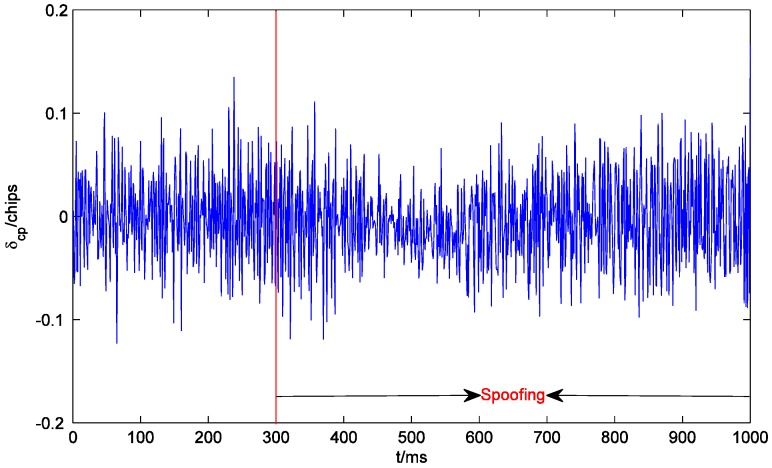
The $\delta_{\mathrm{cp}}$ of the code phase discriminator of PRN21.

**Figure 11 sensors-19-00037-f011:**
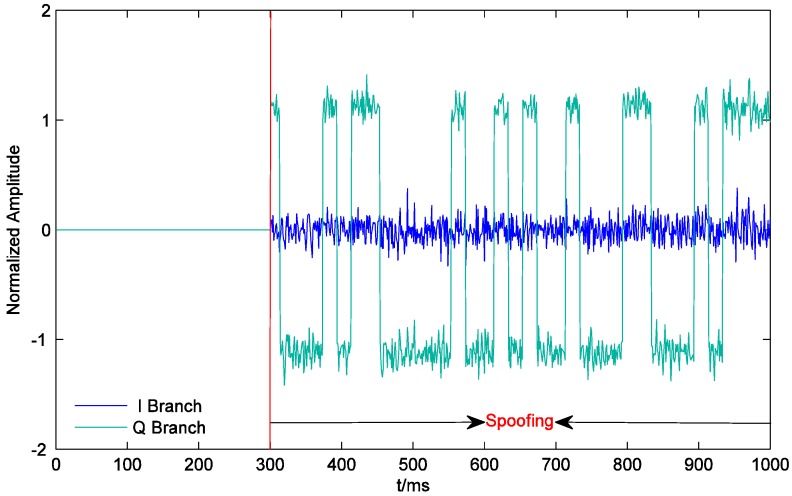
$\hat{x}$ and $\hat{y}$ of the counterfeit signal of PRN22.

**Figure 12 sensors-19-00037-f012:**
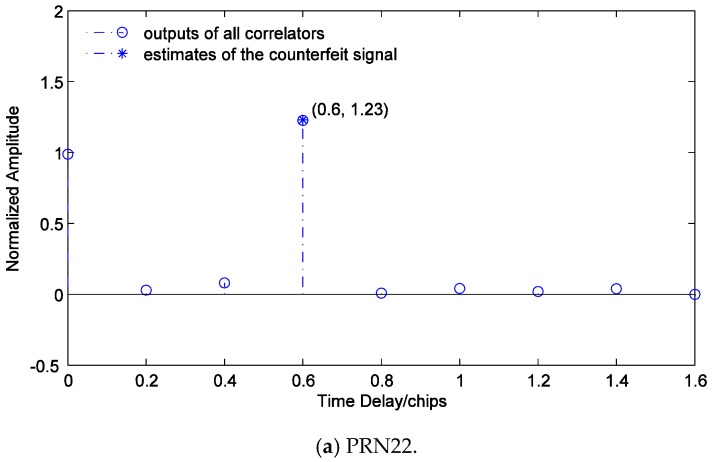
Estimated time delays and amplitudes of the counterfeit signals with short delays

**Figure 13 sensors-19-00037-f013:**
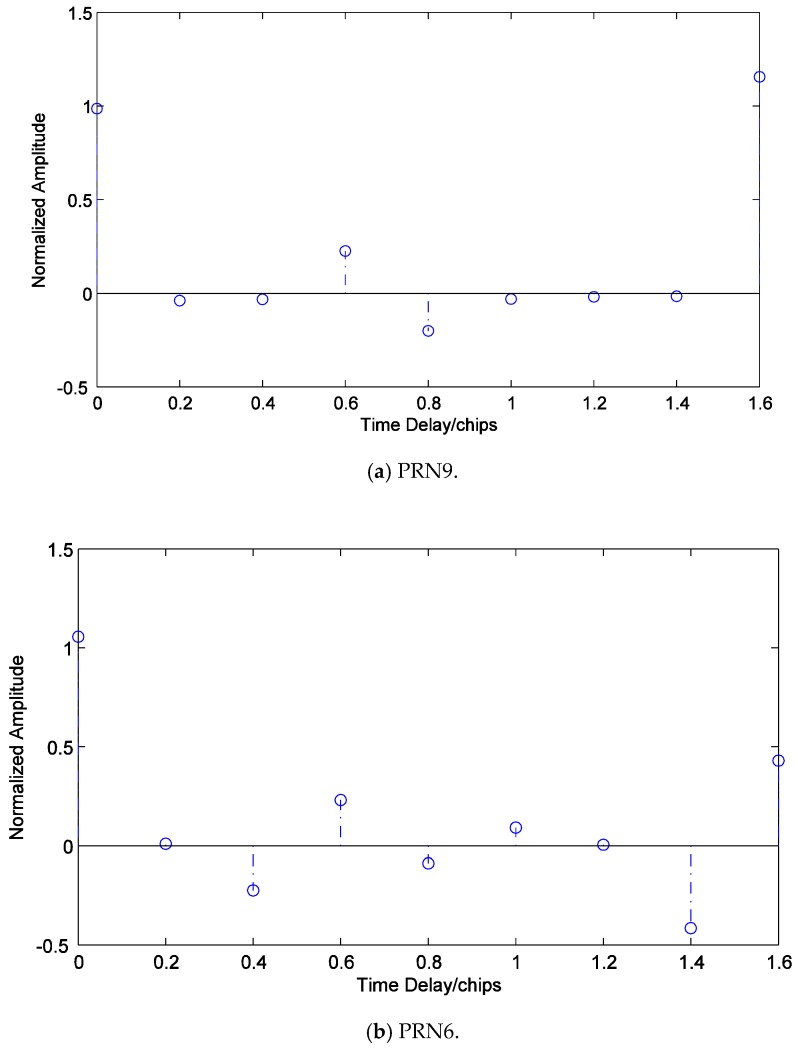
All correlators’ outputs of I branches of the counterfeit signals with long delays.

**Figure 14 sensors-19-00037-f014:**
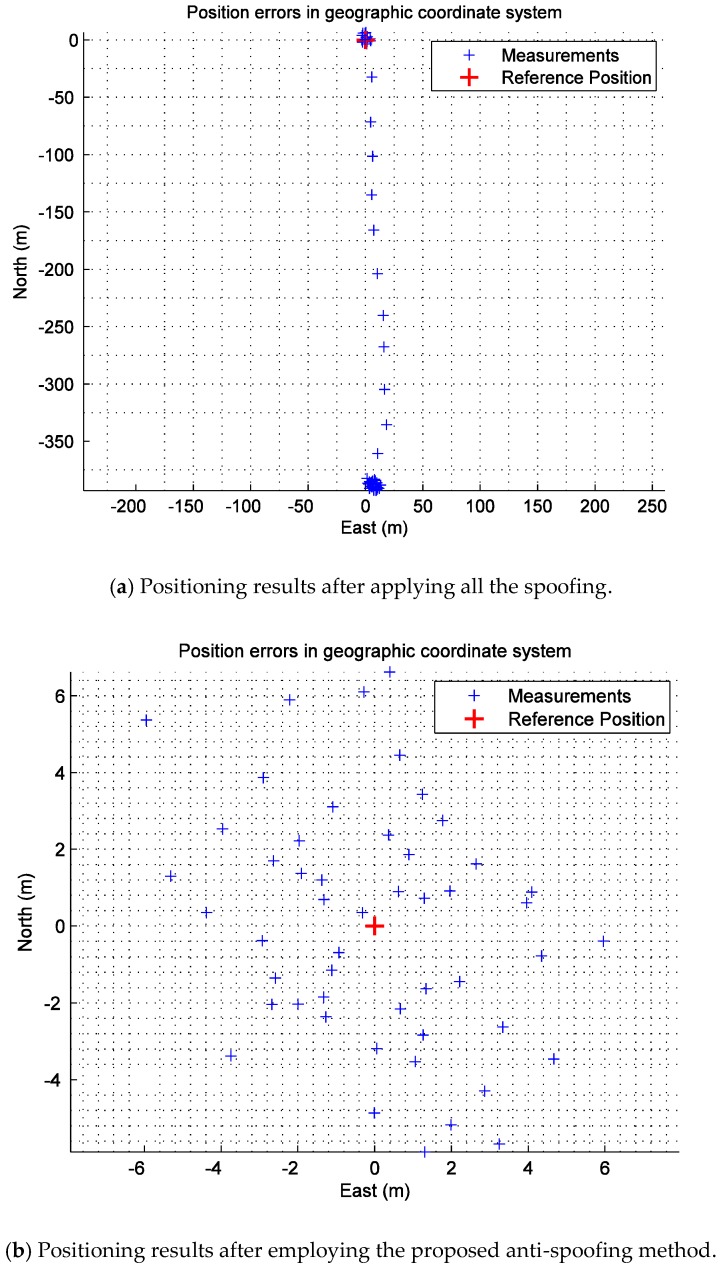
The positioning results after applying all the spoofing and the positioning results after employing the proposed anti-spoofing method.

**Table 1 sensors-19-00037-t001:** Time delays of counterfeit signals.

Counterfeit Signal	Fixed Delay
PRN18	0.11 chips
PRN15	1.05 chips
PRN26	1.5 chips
PRN9	1.69 chips
PRN6	2.49 chips

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
