# Peer review of "Spoofing Detection and Mitigation in a Multi-correlator GPS Receiver Based on the Maximum Likelihood Principle"

_sensors, 2018, doi:10.3390/s19010037_

Round 1
Reviewer 1 Report
General comments:
- The research methodology used in the paper for the experiment is questionable. The presented spoofing scenario is a made up version, and not really representative of usual intuitive spoofing scenarios that can be seen in other researches. I would recommend the researchers to use University of Texas’s TEXBAT spoofing data sets that can be freely downloadable from the following site: https://radionavlab.ae.utexas.edu/index.php?option=com_content&view=article&id=289:texas-spoofing-test-battery-texbat&catid=50
At a minimum, it is recommended that a sophisticated spoofing scenario should be used.
- The presentation of the overall approach ‘Detection and removal of counterfeit signals’ are not at all well-presented, and therefore, it is way too complicated to understand even for a GNSS expert.
- Selection of parameters, i.e., number of correlators, spacing of multi-correlators (i.e., 0.2 chips) are not well justified in the context of the research.
- Usually, most modern receivers now-a-days use narrow correlator or more advanced discriminators for tracking, whereas in the presented work, a wide correlator is used with 0.5 chips spacing between prompt and late correlators.
- What is really interesting is to see that the authors did not really offer any detail on how they reconstruct the counterfeit signal, and then finally removed it from the composite signal. It is one of the most dangerous operation that anyone would attempt to do, and unfortunately there is no clear explanation on the removal of counterfeit signal.
- Tracking is a continuous process, and therefore, it is not enough substance to only show that you estimate your counterfeit signal parameters for only one instant. This is something which is changing over time in a realistic situation, and you keep track of your counterfeiting signal for successive epochs. The claims and the plots presented in the manuscript unfortunately are not convincing enough to hold the claims made by the authors.
- There are lot of assumptions made without much judgmental clarifications, which is also not well understood.
With all the above uncertainties, as a reviewer, I would recommend a ‘reject’ for this manuscript.
Author Response
Details refer to the attachment

Reviewer 2 Report
Dear Editor,
The article “Spoofing Detection and Mitigation in a Multi-correlator GPS Receiver Based on the Maximum” by Yanbing Guo, Lingjuan Miao and Xi Zhang is devoted to important issue – safety of navigation. The authors suggested a novel approach to not only reveal spoofing attack but also to recover authentic signal based on multi-correlator. I should note that authors also suggested an approach to lock signal under several spoofing signals. In my opinion the article lacks only results on direct positioning (or noise in phase and pseudo range). Spoofing signals might increase noise level (even after removing), that can deteriorate precision of estimated coordinates (similar to solar radio emission [Demyanov et al., 2013]). I would recommend to include in the introduction the article [Jafarnia-Jahromi et al., 2012]. I would also recommend to mark in the figures in gray all time domains where there are spoofing signals. The English is understandable and I think that article can be published in the journal.
References:
1) V.V. Demyanov, Yu.V. Yasyukevich, S. Jin (2013). Effects of Solar Radio Emission and Ionospheric Irregularities on GPS/GLONASS Performance, Chapter in Book Geodetic Sciences - Observations, Modeling and Applications, Prof. Shuanggen Jin (Ed.), P. 177-222, ISBN: 978-953-51-1144-3, InTech, 2013 Available from: http://www.intechopen.com/books/geodetic-sciences-observations-modeling-and-applications/effects-of-solar-radio-emission-and-ionospheric-irregularities-on-gps-glonass-performance.
2) Ali Jafarnia-Jahromi, Ali Broumandan, John Nielsen, and Gérard Lachapelle, “GPS Vulnerability to Spoofing Threats and a Review of Antispoofing Techniques,” International Journal of Navigation and Observation, vol. 2012, Article ID 127072, 16 pages, 2012. https://doi.org/10.1155/2012/127072.
Author Response
Details refer to the attachment

Reviewer 3 Report
The manuscript proposes a method to detect and cancel spoofing signals in a receiver that implements a multi-correlator structure. The proposed method is based on an idea already developed for multipath estimation and mitigation, here extended to spoofing signals.
In the reviewer’s opinion, the paper is well written and well organized in general terms. The assumptions, methodology, test setup and results are clearly identified and well explained.
However, some basic assumptions (discussed here below) appear definitely weak and severely undermine the effectiveness and significance of the proposed algorithm. The authors must address the issue and propose their solution, or clarify where the reviewer misunderstands.
The multi-correlator scheme shown in Figure 3 employs multiple copies of the local code at different delay, within and beyond one chip period. However, such a scheme is useful to identify multiple correlation peaks in the hypothesis that the Doppler frequencies of different signal copies are very close, as well as their Doppler rates. This is because the local code generator must align the local code rate with the code rate of the incoming signal, otherwise the output of the coherent integration is extremely degraded or even meaningless. The carrier tracking issue is not explicitly addressed in the manuscript: what appears is that the carrier tracking loop is locked to the authentic signal. In the signal generation setup used in the manuscript (shown in Figure 5) the hypothesis of EQUAL carrier signal for the authentic and counterfeit signal is obviously met, but in the real life it is hard to assume that the spoofer generates each counterfeit signal with the same carrier frequency of each authentic signal (or sufficiently close each other). The problem is that the capability of the scheme in Figure 3 in such real-life situations is really questionable.
Another aspect of this apparent flaw is in Step 5 on page 7: The manuscript claims that the counterfeit signal can be reconstructed (?) and added with opposite phase to the composite signal for removing the original counterfeit component. But this operation is effective only if the delay, frequency, phase and amplitude of this signal component are precisely known (enough), otherwise cancelation is partial or marginal: if the frequency is wrong, this operation could be even disadvantageous. Furthermore, the operation tan(phi) = y^/x^ is known to give very noisy results in practice (this is the reason for which carrier tracking loops employ loop filters): it is weird to directly use it for signal reconstruction and cancelation.
Other minor aspects and comments:
· In (1) and (2) the letter n is used to indicate both the index of the summation and the additive noise: the notation is confusing
· In (1) and (2) why assuming N different counterfeit signals? By the way, from the next section on, the assumption changes in N=1.
· Line 154: Under which hypothesis, apart from a simulated situation, can you assume N and tau_n known? It is very strong claim
· Line 182: the sentence “Accordingly, this will adversely affect…” is not clear: why long delay signals are so detrimental for the procedure?
· In section 4, it is not clear how the authentic signal is identified as “authentic”: how the algorithm guarantees that the canceled signal is not the authentic one?
· Line 194-5: how A can be set? How can you assume that the C/N0 measurement is reliable in the presence of spoofing signals? C/N0 is known to be quite sensitive to the presence of spoofing signals.
· Line 219: It is not clear how it is possible “eliminating the effect of the authentic signal on the estimated amplitude of the first correlator”
· Line 250-1: the idea of moving the antenna along any small trajectory is valid, but of course it requires that the receiver is in movement “at the initial tracking stage”: is it your assumption? In which situation is it valid?
· Section 5: the significance of the results is hindered by the issue highlighted before, confirmed in line 312-3 (“The counterfeit … same carrier phase”).
· Figure 12: at which time instant are the plots referred to? It is not indicated.
· Line 350: the sentence “several anti-spoofing methods are reviewed” is far too pretentious. There is a very quick literature overview in the introduction, then the focus is moved on the authors’ proposal only. No comparison with “several anti-spoofing methods” is proposed. Review papers are different.
· Line 385: An unpublished internal memorandum (internal to which institution?) should not be used as reference, unless unavoidable. But this does not seem the case. A replacing for reference 5 should be proposed.
Author Response
Details refer to the attachment

Round 2
Reviewer 3 Report
In this review the authors address in detail all the comments raised with explanations and justifications in the cover letter. However, sometimes the clarification is not reflected enough in the manuscript, which still leaves some doubt in the reader.
· Response to comment 1 – Reply:
The authors explain clearly their view in the response. Although I am not fully convinced, the explanation is consistent. However, the discussion about the type of attack (locked frequency) must be moved in the introduction, in order to clarify from the beginning the scope adopted for the proposed method. Furthermore, the (strong!) assumption about the high C/N0 ratio for using the ‘tan’ relationship must be clearly declared before the algorithm description.
· Response to comment 3 – Reply:
The provided explanation has not been included in the manuscript: it should be at the beginning of section 3 for clarity.
· Response to comment 4 – Reply:
It is my opinion that also N does not comply with your assumption, not only \tau_n.
· Response to comment 6 – Reply:
The provided explanation has not been included in the manuscript: it should be for clarity.
Overall, I remain highly doubtful about the effectiveness of the method in more realistic situations than in lab simulations; in particular, my major concern is the behavior of the algorithm when the assumptions are not met. I’d suggest the authors to try to elaborate on the boundaries of the scope of their algorithm and to discuss how to manage the out-of-scope situations.
Author Response
Details refer to the attachment
